# Provide energy-aware routing protocol in wireless sensor networks using bacterial foraging optimization algorithm and mobile sink

Shayesteh Tabatabaei 🔟 *

Department of Computer Engineering, Higher Education Complex of Saravan, Saravan, Iran

* shtabatabaey@yahoo.com

## Abstract

Wireless sensor networks (WSNs) include small sensor nodes with battery and processing power and limited memory units then improving power consumption is a major design challenge for any sensor network. In this paper, a new algorithm for routing in the wireless sensor network is proposed using the ultra-innovative algorithm for bacterial Foraging and mobile sink, which leads to energy efficiency. In the proposed method, the number of sensor nodes is determined according to two criteria: the amount of energy on the battery surface and the distance to the sink ahead, which leads to the formation of regular clusters in the network. Nodes adopt a multi-step routing scheme within the network to communicate with the sink. also, the mobile sink is used to balance the load and help consume uniform energy throughout the network. The simulation results show better performance of the proposed method in terms of energy consumption by 17.99%, throughput rate by 30.04%, end-to-end delay by 46.04%, signal-to-noise ratio by 32.81%, delivery rate successfully Data to the sink is 0.80 times higher than the AFSRP (Artificial Fish Swarm Routing Protocol).

**Data Availability Statement:** All relevant data are within the manuscript and its Supporting information files. Public data set: https://figshare.com/articles/dataset/Data_sets_rar/17430710

## 1. Introduction

Wireless sensor networks (WSNs) include small sensor nodes with battery and processing power and limited memory units. There are two important features of WSNs, one is infrastructure-free and the other is self-organizing [1]. These features make sensors a good choice for multiple applications. However, sensor networks have sensor nodes with limited battery power, so how to save node energy while maintaining proper network performance is a challenging issue in designing wireless sensor network protocols.

Deploying WSNs is easier and faster than wired sensor networks or other wireless networks [2] because they have no fixed infrastructure [3]. Because sensor nodes are often used densely, they can withstand network error. Wireless sensor networks do not require a central organization and are configured on their own [2]. There are several types of wireless sensors, such as seismic, low-rate magnetic, thermal, and visual, infrared, acoustic, and radar sensors. These

Public DOI for Data set:10.6084/m9.figshare. 17430710.

**Funding:** The authors received no specific funding for this work.

**Competing interests:** The authors have declared that no competing interests exist.

sensor nodes can monitor a variety of environmental conditions. Some of these conditions include temperature, pressure, humidity, soil arrangement, vehicle motion, noise level, lighting conditions, the presence or absence of certain types of objects, and levels of mechanical stress in attached objects [4]. Wireless sensor networks have different types of applications. The main applications of WSN are military, environmental applications, healthcare applications, home applications, and other commercial applications [5].

In wireless sensor networks, each node receives a signal from a limited area. This signal is processed at that sensor node, and post-measurement information is generally transmitted to observers (for example, base stations). Sensor nodes consume energy when receiving information, processing information, and transmitting the information. In most cases, these sensor nodes are equipped with non-rechargeable batteries. Therefore, energy efficiency is a major design goal in wireless sensor networks [6].

Clustering is one of the ways to improve energy consumption. In clustering, nodes can be divided into several small groups (called clusters) to aggregate data through an efficient network organization [6]. In general, each cluster has a cluster head that coordinates the data collection and aggregation process in a particular cluster. Each member of the cluster transmits its data packets to the cluster head. Clustering in wireless sensor networks ensures a fundamental performance achievement even with a large number of sensor nodes [7]. In other words, clustering improves the scalability of wireless sensor networks, thus minimizing the need for a central organization and promoting local decisions. Each sensor network can perform its tasks as long as it is considered live.

This network has a large number of sensor nodes that are responsible for receiving information from the environment and sending information to a base station or sink in a single step or multiple steps. Nodes close to the sink usually drain their battery resources due to the sharing of multi-step paths and the concentration of data towards the sink more than other nodes. It can be disconnected from the network and the data generated by the sensors can no longer be accessed. The use of portable sinks has been suggested as a possible solution to this problem. The dense area(hotspots) around the sink changes with the mobility of the sink, and the increase in excessive energy consumption around the sink spreads throughout the network, which helps to achieve uniform energy consumption.

Accordingly, in this paper, a new energy-efficient clustering algorithm based on the meta-heuristic algorithm of bacterial foraging and mobile sink for wireless sensor networks is presented which can cluster the network nodes based on the energy level of the nodes and the distance to the sink. Although researchers have proposed many clustering algorithms for WSN there are still many shortcomings in the WSNs in terms of CH selection or finding the best trajectory length. In most studies, CHs that are selected in the first round collect data. However, the CH set to be selected for all rounds and the detection of the location of each CH will facilitate the function of the mobile sink node [8]. Also, it will be more reasonable to construct paths that occur when the mobile sink travels CHs in the shortest manner. This paper introduced a novel method with a BFO algorithm and mobile sink which maintains a good balance between the computational and communication demands of a sensor node in the wireless sensor network. We proposed a Homogeneous WSN with a mobile sink in order to minimize energy consumption in WSN that develops a CH selection algorithm based on the bacterial foraging paradigm. CHs are dynamically selected from the advanced nodes according to the proposed algorithm. This method differs from the existing studies. CH selection is made with the greedy approach that is based on the interaction fitness value, energy node degree, and distance to sink. The best trajectories, thanks to intersection edge points of the visited CHs, are obtained. In this way, the mobile sinks entry to transmission range boundaries of the CH will be a sufficient strategy to collect information. in this paper, we have implemented

comprehensive simulation analyzes by measuring lots of performance metrics, including energy consumption, throughput rate, end-to-end delay, signal-to-noise ratio, the delivery rate that we introduced.

## 1.1 Research contribution

The novelty of this paper lies in the following contributions done:

1. Centralized clustering: Centralized clustering using the bacterial foraging algorithm is executed by the base station i.e. sink. Since the sink is a node equipped with unlimited power so this results in conserving the energy. Moreover, centralized clustering also helps in improving the network scalability as most of the data communication took place inside the clusters.

2. Node's fitness calculation is based on multi-criteria: The bacterial foraging algorithm is practised to select the CH node with the highest fitness value, based on two criteria such energy node degree, and distance to sink. The fitness of a node is evaluated to select the best node among the nodes in the cluster.

3. Head node selection using bacterial foraging optimization approach: The bacterial foraging algorithm is practised to select the CH node for the current round. also in the CH nodes near the sink, data transfer from the CHs to the sink node is done by selecting intermediate nodes as the next-hop. These intermediate nodes are the nodes that are having the least distance from the sink node.

4. Mobile sink: In the CH nodes farther from the sink, the sink moves toward them to Receive the data then this method could conserve the energy and increase the lifetime of the network.

5. Multihop communication among sensor nodes: Data transfer from the CHs to the sink node is done by selecting intermediate nodes as the next-hop. These intermediate nodes are the nodes that are having the least distance from the sink node.

6. Simulation Outcomes: The performance of BFOABMS is investigated by considering different performance metrics like energy consumption, End to End delay, media access delay, packet error rate, throughput, signal to noise rate and delivery rate successfully and it is proved that BFOABMS enhances the energy consumption by 17.99%, throughput rate by 30.04%, end-to-end delay by 46.04%, signal-to-noise ratio by 32.81%, delivery rate successfully Data to the sink by 0.80 in comparison to the AFSRP protocol.

The sections of this article are organized as follows: The second section describes related work on routing protocols on mobile case networks in the third section, the proposed method is stated in the fourth part of the simulation, the proposed method and simulation results in OPNET (Optimized Network Engineering Tools) simulator are given, In the fifth section, the conclusion is stated.

## 2. Related works

In the paper [9], a reliable routing protocol based on clustering and mobile synchronization in the wireless sensor network was presented, which increases the lifespan of the network by using mobile synchronization and clustering. It also increases the reliability of the network by deciding on the best alternative cluster head in the event of an error, locally and dynamically. The proposed protocol, called DCRRP (Distributed Clustering Reliable Routing Protocol), is distributed and can minimize data report latency. The results of the simulated protocol proposed in OPNET are compared with the NODIC (Novel Distributed Clustering) protocol in error and error-free mode. In general, it was observed that the method proposed by the

authors shows better behaviour than the NODIC protocol. The disadvantage of this method is that after moving the sink, several nodes remain as sinks in the original location of the sink, and the nodes must, if present, inform the nodes representing the mobile sink, which in itself will cause energy consumption.

In this paper [10], an evolutionary method based on clustering using an optimized invasive weed algorithm based on fuzzy modelling for WSNs is proposed. Which can select the most suitable node as the cluster head to increase the life of the network and also address the challenge of energy conservation in sensor networks. The authors used a fuzzy inference model to evaluate the fit of each node in the network. The simulation results show a significant reduction in the number of dead nodes in each run and a reduction in energy consumption in the sensors. In addition, their proposed method increases the network stability period by up to 45% compared to the artificial bee protocol and 18% compared to the artificial bee quantum protocol. The advantage of this method is the use of an optimized aggressive weed algorithm that can adapt and be random and can quickly find the global optimal [11], as well as the use of a fuzzy inference model that has high accuracy. The disadvantage of this method is the high processing cost due to the use of fuzzy logic with an aggressive weed algorithm.

The paper [12] presents a centralized method for improving energy efficiency with fuzzy-based clustering and machine learning called ECFU (Energy-based Clustering with Fuzzified Updates) for wireless sensor networks. The proposed method solves the problem of improving the lives of the network and at the same time balancing the energy between the sensor nodes of the network. To achieve greater efficiency in network operations, a clustering technique based on fuzzy updating and machine learning has been used. Clustering is a useful way to improve power consumption to extend the life of wireless sensor networks. Exchanging messages due to repeated recreation burdens the sensor nodes and wastes battery power. This paper presents a modified clustering method that reduces the overhead of clustering and message exchange and thus performs the clustering operation effectively. The network is clustered according to the residual energy of the sensor nodes. To increase the network life, the proposed algorithm calculates the next update cycle using a fuzzy inference system. Fuzzy criteria, i.e. node residual energy, distance from the sink, and average data rate, are used to calculate the cluster update cycle. Nodes apply machine learning clusters at regular intervals to classify data based on their similarity. Classified data is transferred to the cluster head after reducing the number of message transmissions. The proposed method improves the energy consumption of clustering and data transmission. The proposed method reduces clustering overhead, which is considered a major challenge in the implementation of network clustering protocols.

In [13] the authors use a multi-network and distributed network routing scheme to increase network failure tolerance and conserve sensor energy. The simulation results showed that the sub-path detection method represents an efficient method that is well consistent with the properties of the WSN multidimensional communication model. The main idea of this type of routing is to tag messages related to each route along with identifying adjacent routes that send the message.

In [14], the authors proposed an optimal scheduling mechanism and method for sending packets so that information is collected at the cluster heads in wireless sensor networks. The purpose of the method is to investigate real-life process processes for a specific time interval in routing packets with the least probability of losing the packet to the base station. To achieve this, they improved the scheduling algorithm to arrange time slots through which data must be sent by sensor nodes. The scheduling method ensures that all packets are sent in a specified slot and thus the delay is reduced and there is also a possible packing probability for each node.

In [15] the authors provide an analytical model for network longevity and coverage problems in wireless sensor networks using a two-dimensional Gaussian distribution. The

probability of coverage in the Gaussian distribution depends on factors such as the standard Gaussian deviation, the distance between the target point and the centre point. By adjusting the various parameters mentioned above, the probability of occurrence of coverage and network life can be increased. Using the timed node deployment algorithm, longer network life and larger coverage are achieved using a limited number of sensor nodes.

In the paper [16], the authors proposed a method called FMSFLA(Fuzzy Multi-hop Frog Leaping Algorithm) to maximize network life and scalability based on the application, which considers the effective parameters for selecting the cluster head and parent node. In the proposed method, the jumping frog algorithm is used to provide a fuzzy multi-step clustering protocol (FMSFLA). SFLA (Shuffled Frog Leaping Algorithm) is used to automatically configure and optimize the base rule table in a fuzzy inference system with five parameters that can be adjusted in two steps, namely cluster head selection (CH) and parent selection, based on application features. The proposed protocol (FMSFLA) takes into account the effective parameters such as energy, distance from the base station (BS), number of adjacent nodes, and actual distance of the node from BS, average path load, latency, overlap, and hotspot problem. The proposed method is expected to have the best performance in each round of execution. In each round, the steps of CH selection, parent selection, cluster formation, and steady-state are performed. In the CH selection step, CH is selected from the candidate nodes based on the fuzzy output and energy threshold (i.e. a control parameter) and according to the overlap rate of adjacent CHs. In the proposed protocol, the parent selection phase begins with determining the level of CHs in the network. At the end of this step, the parents of each CH are determined based on the maximum fuzzy output based on the application. In the cluster formation stage, clusters are formed based on the specified CHS. Finally, the information received by the CHs is sent to the BS through their parents in a stable phase. FMSFLA evaluates received packets based on the number of live node protocols. The proposed Fuzzy Multi-Step Clustering Protocol (FMSFLA) is not only energy efficient but can also manage the energy consumption of nodes based on the FIS (Fuzzy Inference System) rule table and five control parameters using frog jump. The proposed protocol also considers the effective parameters such as energy, distance to BS, number of adjacent nodes, actual distance between a node and BS, average path load, latency, overlap, and hotspot problem to achieve the best. According to the simulation results, FMSFLA performs better than other protocols in all scenarios. The disadvantage of this method is the high processing cost of the jumping frog algorithm and fuzzy logic.

In the paper [17] a grid-based routing algorithm is proposed for the sensor network to conserve energy in the sensor nodes and increase the network life. In addition, routing is done through a network coordinator that applies fuzzy rules to find the optimal path to reduce the number of coordinators in the routing process. In an energy-efficient grid-based routing algorithm, each grid selects its grid using the residual energy of the nodes in the network and separating the nodes from the sink using a fuzzy logic coordinator. Therefore, energy consumption is reduced in sensor nodes. Because the transfer of information from the source to the sink is done by the grid coordinator, which acts as a relay node, it reduces energy consumption in the routing process, which leads to an increase in network life. In this method, the design is done in such a way that the problem of energy consumption in most sensor nodes is used more effectively to prevent energy loss during transmission. To perform effective routing through cluster head nodes, clustering techniques have been used to help save energy consumption in sensor nodes. In addition, fuzzy rules are applied in this work to make effective decisions according to the criteria of node location, energy analysis, cluster formation, cluster selection, and routing through clusters.

In the paper [18] a new concept for mobile sensor nodes called the Cluster-Based Energy-Based Routing Protocol (MEACBM) for routing in a network of hierarchically heterogeneous

WSNs is proposed that selects CH based on the proposed probability equation. Selects only the sensor as the cluster head (CH) that has the most energy in the equation among the other sensor nodes. Therefore, in MEACBM, only sensor nodes with the highest energy compared to low-power sensor nodes are selected as CH. Hierarchical clustering is considered hierarchical for the three levels of sensor nodes that are connected by cluster communication and sensor nodes throughout the network area. In MEACBM, after the establishment of sensor nodes and the formation of clusters, the entire network area is divided into sections and inside each section is placed a mobile sensor node which acts as a collector of mobile data to collect data from CH Acts. This method helps to significantly reduce the energy consumption of sensor nodes to transmit information to the base station (BS). The simulation results show the effectiveness of MEACBM(Mobile Energy Aware Cluster Based Multi-hop) routing protocols compared to other new cluster-based routing protocols in terms of network life, stability, throughput, number of CHs, and the number of dead nodes. The problem with the proposed method is that the selection of the cluster-head is done only according to the energy criterion of the sensor nodes, and on the other hand, the mobile node consumes a lot of battery energy than other nodes and may consume its energy faster and the node will shut down.

In the paper [19], a new method for optimal clustering in circular(circular) networks OCCN (Optimal Clustering in Circular Networks) is proposed, which aims to reduce energy consumption and increase the life of wireless sensor networks. In this method, which is presented for a circular area around a sink, a single-step connection between the eaves and the sink is replaced by an optimal multi-step connection. In addition, the optimal number of clusters is calculated and the energy consumption is optimally distributed by dividing the grid approximately equally. In this paper, a multi-step communication protocol is presented that uses optimal parameters. Optimal parameters such as optimization of one-step communications, an optimal number of cluster sizes, and the optimal number of clusters are considered for a circular network in which the sink is located in the centre of the network. During the clustering operation, the selection of the proposed cluster head method significantly improves the life of the network and the energy consumption process is linear and at least 50% of the nodes survive.

In the paper [20], a reliable routing technique using new delay energy is used to transfer data in a heterogeneous sensor environment. In the proposed method, a limited search space is defined to ensure the timely delivery of sensitive delay data. In addition, in the defined search space, an algorithm has been developed to select a balanced path of energy-delay between the source and the sink, which enables fast data delivery through an energy-efficient step. The proposed method also improves the success rate of data packets received in the sink in a large dense network by collecting data and providing a suitable load balance in the network. The proposed method achieves a good balance between energy consumption and end-to-end delay by selecting the minimum energy of the next node in the long-distance data transfer phase. This method not only solves the hotspot problem in mobile hierarchical routing protocols but also reduces end-to-end delivery delays. In addition, proper load balancing across the network significantly improves the percentage of data packets successfully received in the sink by selecting multiple agent nodes-(Agent Node) and accumulating data during the data transfer phase and it also reduces the overall energy consumption of the grid.

In the paper [21], a new approach to address the problem of energy efficiency in wireless sensor networks is presented by an energy-aware routing protocol using a fish swarm optimization algorithm called AFSRP in these networks, which improves energy consumption. The proposed protocol was simulated with ERA(Energy-aware Routing Algorithm) protocol in OPNET11.5 simulator and the simulation results showed that their proposed protocol performed better than ERA protocol in terms of power consumption, end-to-end latency, media

access delay, bandwidth, transmission rate, probability of successful sending to the sink, and the signal to noise ratio. The problem of this method is that the optimization of fish is used in static environments and for dynamic environments such as sensor networks, the convergence speed may be reduced.

In the paper [22], a re-evolution of the stable improved LEACH (SILEACH) algorithm was proposed to overcome the problems of the LEACH algorithm. SILEACH has two stages, the setup and the steady stage. In the setup stage, a random number between (0–1) is generated in each node and the threshold value is calculated in each node. To select CH nodes, each node, that has the highest RE and less distance to BS, will get more opportunity to act as CH. the steady stage the nodes sense the environment and send the data to the BS via CH. These operations are repeated for each round. This is performed by assuming the BS inside the network area. The simulation results showed that the SILEACH significantly outperforms the Enhanced LEACH (LEACH) and Improved LEACH (ILEACH) protocols in terms of the stability period and network lifetime.

In the paper [23], to overcome limitations of the lifetime and its energy consumption of sensor nodes enhanced the LEACH (low energy adaptive clustering hierarchy) protocol was extended by identifying a cluster head according to the lowest degree of distance from the sink. The proposed method is the same steps as LEACH but a difference occurs in the setup stage in which, in the setup stage, all the regular nodes select a random number between (0–1), then if that number is less than or equal to the threshold the node becomes a CH; member nodes select a CH according to distances to reach the base station, not a CH that is closest to it. Simulation results showed enhanced LEACH can increase the network lifetime and minimize the consumption of energy. The summary and the highlights and limitations of each algorithm in related work showed in Table 1.

## 3. Suggested method

The BIFOA(Bacterial foraging Optimization Algorithm) developed by Passino in 2002 is inspired by the social feeding behaviour of Escherichia coli [24]. Bacteria find food in a way that maximizes the energy produced per unit of time. Bacteria also communicate with other bacteria by sending signals. The process in which bacteria look for food by taking small steps is called chemotaxis, and the main idea of the bacterial foraging optimization algorithm is the chemotaxis movement of artificial bacteria into the problem space. Bacteria move during feeding through a set of expandable flagella. The flagella help E.coli to swim or hang. These are the two main actions when foraging bacteria. When the bacteria have eaten enough food, they increase in length and split in two at the right temperature to form a capsule just like themselves. Passino used the same idea for the reproduction process in the bacterial foraging optimization algorithm. Due to sudden environmental changes or seizures, the chemotactic process may be disrupted and some bacteria may be transferred to other areas this condition in a population of true bacteria can cause all bacteria to be killed or spread to a new part of the environment. Now suppose we want to find the minimum $J(\theta)$ to $\theta = R^p$ (i.e. $\theta$ is the vector P after real numbers) and we have a criterion for measuring or describing the gradient $\nabla J(\theta)$ does not have. The bacterial foraging optimization algorithm follows four mechanisms in the real bacterial system to solve non-gradient problems: chemotaxis mass movement, reproduction, and Elimination and dispersal(Elimination and Dispersal). Symptoms of the four mechanisms listed are:

$\theta$: the vector P after real numbers

J: stage indicator of chemotaxis

K: Reproductive stages index

Table 1. The summary of all algorithms in related work.

| Designer's name | Methodology | Advantages | Disadvantages |
|---|---|---|---|
| [9] | A reliable routing protocol based on clustering and mobile synchronization | It can increase the lifespan of the network and minimize data report latency | After moving the sink, several nodes remain as sinks in the original location of the sink, and the nodes must, if present, inform the nodes representing the mobile sink, which in itself will cause energy consumption. |
| [10] | Clustering WSN using an optimized invasive weed algorithm based on fuzzy modelling | Results showed a significant reduction in the number of dead nodes in each run and a reduction in energy consumption in the sensors | High processing cost due to the use of fuzzy logic with an aggressive weed algorithm. |
| [12] | A centralized method for improving energy efficiency with fuzzy-based clustering and machine learning | The proposed method improves the energy consumption of clustering and data transmission and reduces clustering overhead | High processing cost due to the use of fuzzy logic |
| [13] | multi-network and distributed network routing methods are used to tag messages related to each route along with identifying adjacent routes | It can increase network failure tolerance and conserve sensor energy. | Decentralized networks are more expensive and time-consuming to deploy because need to install and configure multiple servers with load balancing and failover capabilities. |
| [14] | A scheduling mechanism to arrange time slots through which data must be sent by sensor nodes | It ensures that all packets are sent in a specified slot and thus the delay is reduced and there is also a possible packing probability for each node. | If time slots are not correctly diagnosed it causes loss of data |
| [15] | An analytical model for network longevity and coverage problems using a two-dimensional Gaussian distribution | It has longer network life and larger coverage | High processing cost due to the use of two-dimensional Gaussian distribution |
| [16] | The shuffled Frog Leaping Algorithm with fuzzy multi-step clustering protocol | It improves energy consumption | This method is the high processing cost of the jumping frog algorithm and fuzzy logic. |
| [17] | A grid-based routing algorithm, that is done through a network coordinator that applies fuzzy rules to find the optimal path. | It can reduce the energy consumption is in sensor nodes. | High processing cost due to the use of fuzzy logic |
| [18] | offer the clustering-based Routing Protocol in a network of hierarchically heterogeneous WSNs based on the energy of nodes | It can improve network life, stability, throughput, number of CHs, and the number of dead nodes. | It selection of the cluster-head is done only according to the energy criterion of the sensor nodes, and on the other hand, the mobile node consumes a lot of battery energy than other nodes and may consume its energy faster and the node will shut down |
| [19] | A new method for optimal clustering in circular networks | It can reduce energy consumption and increase the life of wireless sensor networks. | This method is performed only for a circular area around a sink, and It is not performed for all nodes in WSN network. |
| [20] | A reliable routing technique using new delay energy is used to transfer data in a heterogeneous sensor environment | The proposed method can improve the success rate of data packets received in the sink in a large dense network by collecting data and providing a suitable load balance in the network also this method not only solves the hotspot problem in mobile hierarchical routing protocols but also reduces end-to-end delivery delays and it can reduce the overall energy consumption of the grid. | It is can only perform in heterogeneous sensor environments but in a real environment the most of the time the sensor nodes are homogenous |
| [21] | An energy-aware routing protocol using a fish swarm optimization algorithm | The proposed protocol can improve power consumption, end-to-end latency, media access delay, bandwidth, transmission rate, probability of successful sending to the sink, and the signal to noise ratio. | Using the fish swarm optimization algorithm can reduce the convergence speed. |
| [22] | Re-evolution of the stable improved LEACH algorithm based on two criteria energy remaining and distance to sink | The simulation results showed that the proposed method significantly outperforms in terms of the stability period and network lifetime. | It has considered only two criteria (energy remaining and distance to sink) to enhance the leach protocol but maybe other important criteria such as processing cost have existed |
| [23] | Enhanced the LEACH protocol, using the cluster head selection according to the lowest degree of distance from the sink. | Simulation results showed enhanced LEACH can increase the network lifetime and minimize the consumption of energy. | It has considered only one criterion (lowest degree of distance from the sink) to enhance the leach protocol but maybe other important criteria such as energy remaining have existed |

L: Elimination and Dispersal Event Index

P: Dimensions of the search space

S: The total number of bacteria in a population

$N_c$: number of chemotactic steps

$N_s$: swimming length

$N_{re}$: number of reproductive stages

$N_{ed}$: Number of Elimination and dispersal events

$P_{ed}$: possibility of Elimination and dispersal

C (i): The size of the steps in the random directions determined by the suspension and p(j, k, l) = {$\theta^i$ (j, k, l)\i = 1, 2, . . .., s} indicates the position of each member in the bacterial S population at $j_{th}$ chemotactic stage, $k_{th}$ reproductive stage And $i_{th}$ is the deletion-scattering event. Here J(I, j, K, l) specifies the cost position of the bacterium $i_{th}\theta^i$(j, k, l) ∈ R ^ p. Note that J is used here as a cost. In a population of real bacteria, s can be very large (e.g. s = 109), but P = 3. In simulations performed to solve search problems, a much smaller and fixed population size is used. However, in the bacterial nutrition optimization algorithm, there is a possibility of p> 3, which is used for optimization problems with higher dimensions. In the following, the steps of the bacterial foraging optimization algorithm are described.

## Chemotaxis

This process mimics the movement of E. coli cells in swimming and hanging through the flagella. A biological E. coli bacterium can move in two ways. It can swim in the same direction for a while or stay suspended. And throughout his life, he alternates between the two. Presume $\theta^i$(j, k, l) indicates the location of $i_{th}$ bacterium in $j_{th}$ chemotactic, $k_{th}$ productivity and $i_{th}$ Elimination and dispersal stage. c(i) the size of the steps in the random directions determined by the suspension (units of swimming length). In this case, the movement of bacteria can be shown according to Eq 1 [24]:

$$\theta^i(j + 1, k, l) = \theta^i(j, k, l) + c(i)\frac{\Delta(i)}{\sqrt{\Delta^T(i)\Delta(i)}} \tag{1}$$

Which $\Delta$ represents a vector in a random direction whose elements are between [–1, 1].

- Group movement: One of the interesting behaviours observed in some bacterial species including E. coli and S. Typhimurium is the group movement of coaxial patterns with high bacterial density. Eq 2 [24] represents the intercellular signalling in the E. coli swarm:

$$\begin{aligned} J_{cc}(\theta, p(j, k, l)) &= \sum_{i=1}^{s} J_{cc}(\theta, \theta^i(j, k, l)) \\ &= \sum_{i=1}^{s}[-d_{attractant}exp(-w_{attractant}\sum_{m=1}^{p}(\theta_m - \theta_m^i)^2)] \\ &+ \sum_{i=1}^{s}[h_{repllant}exp(-w_{repllant}\sum_{m=1}^{p}(\theta_m - \theta_m^i)^2)] \end{aligned} \tag{2}$$

$J_{cc}$ ($\theta, p(j, k, l)$) is the value of the objective function, which is subtracted from the real objective function (to maximize) to show changes in the objective function over time. S is the total

number of bacteria, P is the number of optimization variables in each bacterium and $\theta = ([\theta_1, \theta_2, \theta_3, \ldots \theta_p])^T$ is in the next P search domain. In this regard, $w_{repllant}$, $w_{attractant}$, $h_{repllant}$, $d_{attractant}$ are different coefficients that must be selected correctly. (Usually the values $w_{repllant} = 10$, $w_{attractant} = 0.2$, $d_{attractant} = 0.1$, $h_{repllant} = 0.1$ are considered for these coefficients). Paying attention to Eq 2 shows that it is desirable for bacteria to better explore the search space. In other words, the closer one bacterium gets to other bacteria, the more it will strengthen its final fit. (A contractile interest occurs in the bacterial population.) And the farther a bacterium is from other bacteria, the stronger the second part of the upper relation of the final fit of the bacterium (there is an expansionary interest in the bacterial population). However, because the $w_{attractant}$ value is always considered less than the $w_{repllant}$ [24], there is a contractile interest in all bacteria, so that eventually the algorithm converges.

- Reproduction: Weaker bacteria are eventually killed (those for which less fit is calculated by the objective function), and better bacteria are broken down into two bacteria, which are replaced on the spot, which keeps the population size constant.

- Elimination and dispersal: Gradual or sudden changes in the environment in which bacteria live may occur for a variety of reasons, such as a sudden rise in temperature that kills a group of bacteria. Another thing that can happen is that the bacteria move to another area. To simulate such a phenomenon, in a bacterial foraging optimization algorithm, some bacteria are less likely to be killed and a new number to be randomly assigned to new locations [24]. The flowchart of this algorithm is shown in Fig 1.

In the following, we will explain the details of the clustering of wireless sensor networks using the bacterial foraging algorithm. In the proposed method, each sensor node is considered a bacterium. Also, to simplify data routing, a virtual backbone is created from the cluster heads, which are rooted in the sink node, which is the main source of food. This algorithm consists of two phases, the clustering phase, and the data transmission phase.

Phase 1 Clustering: In the clustering phase, the sensors or bacteria are grouped into separate clusters. This phase includes the following steps.

Step 1: The sink first sends a route request message to all sensor nodes (60 sensor nodes) that include the sink ID and physical position of the sink. Each sensor node that receives this message obtains its position via GPS and calculates its distance to the sink according to Eq 3.

$$D_i = \sqrt{(xs - xi)^2 + (ys - yi)^2 + (zs - zi)^2} \tag{3}$$

$D_i$ is the distance to the sink, $(xs, ys, zs)$ the position of the sink, and $(xi, yi, zi)$ the physical position of the $i_{th}$ sensor.

The distance sensor to the sink then sends the amount of remaining battery power and its ID to the sink in response to the sink request in the form of a response to the route request. The sink, after receiving this information, registers them in its table and the centralized process of selecting the best nodes as in-sink cluster heads is performed using a bacterial foraging algorithm. The initial population of sensor node IDs is randomly selected as the cluster head, so that since we want to specify five cluster head nodes for 60 nodes, it creates ten arrays of five that inside each house of arrays is the sensor node IDs that Candidates are randomly inserted, so in each solution, up to five bacteria are inserted. Each sensor acts as a bacterium. Fig 2 shows an example of a solution that randomly selects sensor nodes (bacteria) with identifiers 60, 4, 56, 34, and 45 as cluster head candidates.

The initial population containing up to 10 solutions is randomly generated.

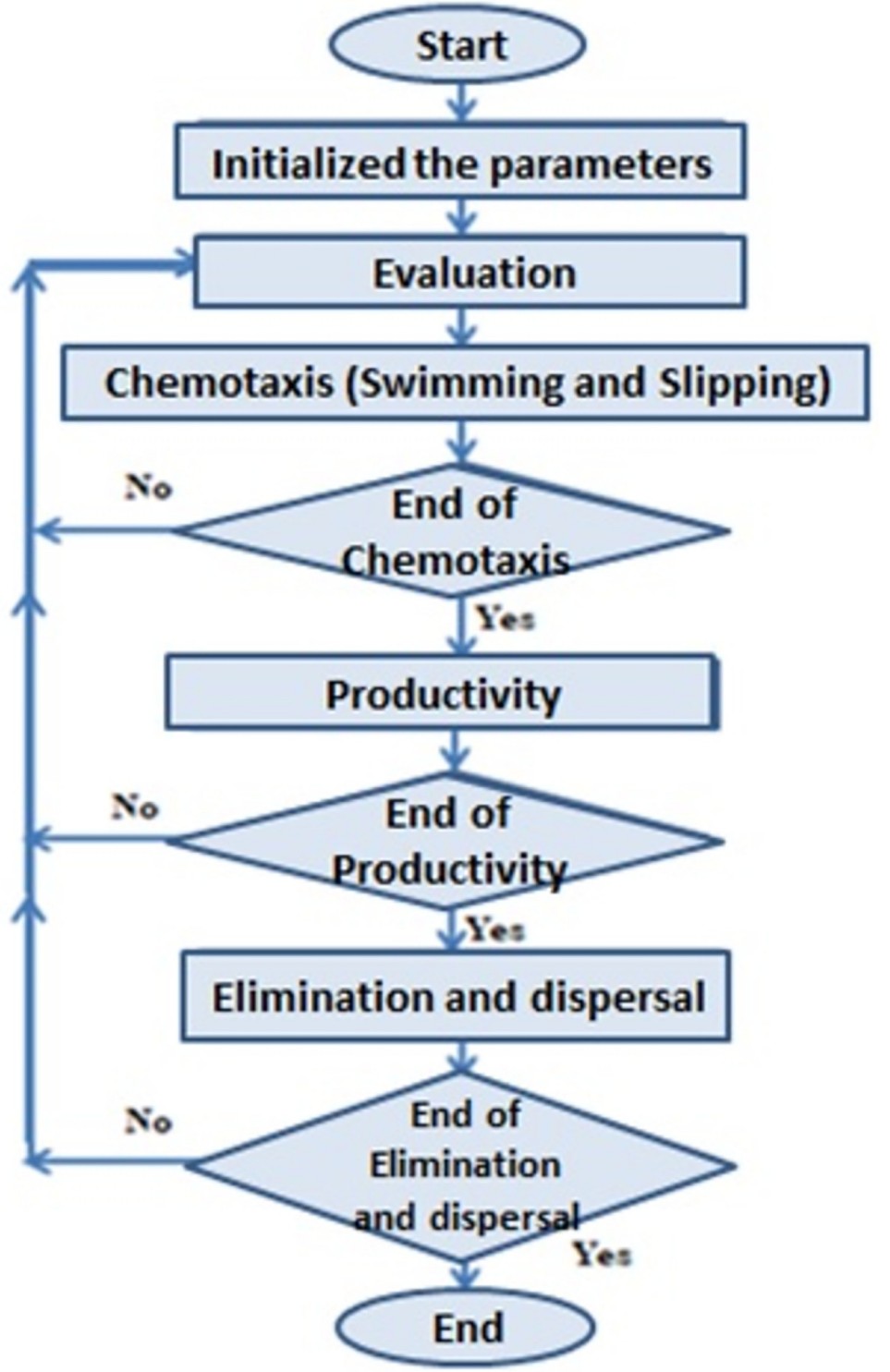

**Fig 1. The flowchart of the algorithm.**

| Cluster head candidates. | 40 | 10 | 6 | 58 | 45 |
|---|---|---|---|---|---|

**Fig 2. An example of a solution in the proposed method.**

Step 2. Chemotaxis is performed at this stage. Chemotaxis is the motor behaviour of bacteria, which is called the life cycle of bacteria. This behaviour involves repetitive Ns (lifetime) in which the bacteria take steps to search for brain material. The values of $w_{repllant} = 10$, $w_{attractant} = 0.2$, $d_{attractant} = 0.1$, $h_{repllant} = 0.1$ are considered and also initializes the values of the variables Ned, Nre, Ns, Nc to 10, 100, 10, and 10, respectively. And $\theta^i(j, k, l)$ is also considered the position of the $j_{th}$ sensor. If the first round is clustering, the movement of each bacterium is calculated according to Eq 1 and the position of the bacterium is updated.

Since bacteria in certain conditions secrete adsorbents that attract other bacteria to a specific area. According to this relation, concerning updating the cost function of each bacterium, after moving it according to Eq 2, the value of $J_{cc}(\theta, p(j, k, l))$ should be added to it, which represents the amount of absorbing and repelling forces between bacteria in the population. Therefore, for the second round, according to Eq 2, the bacterial movement is calculated, which is Eq 2 $(\theta_m - \theta_m^i)$ considers the distance of each bacterium or sensor to the sink and updates the position of the bacterium, i.e after calculating the bacterial movement, the position The new result is checked according to the position **of** which sensor node. The ID of that sensor is placed in the bacterium and the bacterium is updated.

Step 3. At this stage, reproduction takes place. At this stage, after the life cycle of the bacteria to move is over, the health of the bacteria, which is proportional to the number of nutrients accumulated during the life cycle, is calculated based on fitness for all bacteria. Then some of the bacteria with the least fitness function die (are removed) and the same number of the best bacteria multiply. (They become two bacteria.) For each bacterium inside the array, the solution in the initial population calculates the fitness value according to Formula 4.

$$Fit = RE_i + \frac{1}{D_i} \tag{4}$$

As *Fit* is the fitness, *RE* is the remaining energy of the $i_{th}$ sensor node and *D* is the distance of the $i_{th}$ sensor node to the sink. Organizes the bacteria according to their suitability and selects fifty per cent of the bacteria with high suitability and inserts them in the initial population [24].

Step 4. The Elimination and dispersal step is done. In a real swarm of bacteria, the effect of environmental changes such as rising temperatures may kill many bacteria or move them to other areas. Inspired by this behaviour, after a certain number of repetitions of the reproductive stage, each of the bacteria is likely to be removed by $P_{ed}$ and thrown to another place (it becomes another bacterium). For each bacterium selected from the population in step 3, a random number is generated if this number is less than 0.8 of the Elimination and dispersal value ($P_{ed} = 0.8$). A sensor with a random ID generates and replaces the sensor ID in the solution.

Step 5. The population is evaluated according to Eq 5 and sorted according to the fitness function. Selects the best solution from the population that has the highest fitness as the cluster

head.

$$Fit = \sum\nolimits_{i=1}^{5} \left( RE_i + \frac{1}{D_i} \right) / 5 \tag{5}$$

Step 6. The sink sends a notification message of being cluster head, indicating the nodes within the array of the best solution, informing them that they are cluster heads. Each cluster head then generates an Advertisement message that contains information about its physical location and ID and distributes it in its range. They are connected to the nearest cluster head by sending a connection message which contains the ID of the member node and the amount of energy remaining and the physical position of the member node and thus clusters are formed.

The second phase of data transmission: In sensor networks, in cases where the data transmitter is not able to send with one step, it uses multi-step relationships. One way to reduce energy consumption is to use multi-step relationships, to use routes that require less energy to send data for routing. On the other hand, increasing the number of steps causes more nodes to buffer along the closed path, which will have high processing and overhead. Therefore, it is very important to use an asynchronous port that helps to reduce the number of steps in multiple routings in the network. After the data collection is completed by the cluster heads, the cluster head nodes send a message to the sink, informing the sink of their residual energy after a period of message exchange and telling the sink that they have data to send. The sink will make a decision based on the residual energy of the sprigs in its table. If the residual energy of the sprigs is more than 70% of their total initial energy, the sink moves to a randomly installed location, but if it is less than 70%, the sink moves to a denser area and places a node as its representative in its embedded location to be responsible for collecting data from the cluster head [9]. The cluster heads will now be notified by the sink of both specified coordinates to receive the data, and send the data to the nearest one using several steps or a single step. The pseudo-code of the proposed method is presented in Fig 3 and the platform configuration of the proposed method is presented in Fig 4. The proposed method energy consumption model is evaluated according to eruption 6 [25].

$$E_C = argmax \frac{RE_i}{maxRE_i \sqrt{D_i}} \tag{6}$$

As $E_c$ is the energy consumption, $RE$ is the remaining energy of the $i_{th}$ sensor node and $D$ is the distance of the $i_{th}$ sensor node to the sink.

## 4. Simulation of the proposed method

### 4 1 Simulation environment

In this paper, we use Contiki operating system that uses standard C and follows an event-driven programming model and OPNET(Optimized Network Engineering Tool) [27] Modeler version 11.5 simulation software is used to simulate the proposed method and compare it with the AFSRP protocol. The simulation parameters are shown in Table 2. According to Fig 5 in the proposed method, we have considered the network correlation of 60 nodes, of which two scenarios, the first scenario of the sensor nodes are randomly distributed in the environment according to the AFSRP protocol and in the second scenario, the nodes are randomly distributed in the environment that is clustered by the proposed method (bacterial foraging algorithm). The proposed algorithm is named BFOABMS (Bacterial Foraging Optimization Algorithm Based Mobile Sink).

```
Initializing parameters
P, S, Ns, Nre, Ned, Ped, C(i)
Starting Elimination-Dispersal loop with l=1,2,…,Ned
        Starting Reproduction loop with k=1,2,..,Nre
                Starting Chemotaxis loop with i=1,2,…,S
                        Compute the fitness function J(i,j,k,l)
                        Set Jlast=J(i,j,k,l)
                        Tumble. Generate Δ(i)on [-1,1]
                        Move. Compute θ(i,j+1,k,l)

                        Swim:
                                For t=1 to Ns (swim steps)
                                If J(i,j+1,k,l)<Jlast then
                                Update Jlast
                                Update θ
                                Else stop
                Repeat until i=S

                Reproduction
                Compute the health status (Jhealth) for the bacterium i
                Sort the bacteria ascendingly according to Jhealth
                For Sr with the best values of Jhealth do split
        Repeat until k=Nre

        Elimination and Dispersal
                For m=1,2,…,S
                        Eliminate each bacterium with ped<Ped
                Repeat until m=S
Repeat until l=Ned
End
```

**Fig 3. Pseudo-code of the proposed method [26].**

We have assumed the same correlation for both scenarios. The following are the simulation results of the proposed protocol based on the scenarios.

## 4.2 Performance criteria in the proposed method

To evaluate the efficiency of the proposed method, the following criteria are used.

- Energy Consumption: The energy consumption is equal to the sum of the energy used by the nodes within the network, where the energy used for a node is equal to the sum of the energy used for communication, including transmission and transmission, and waiting.

- End-to-End delay: The time it takes for a data packet to be transmitted from sender to receiver. To calculate the average end-to-end latency, the end-to-end latency of all packets received by the receivers is calculated and their average is calculated.

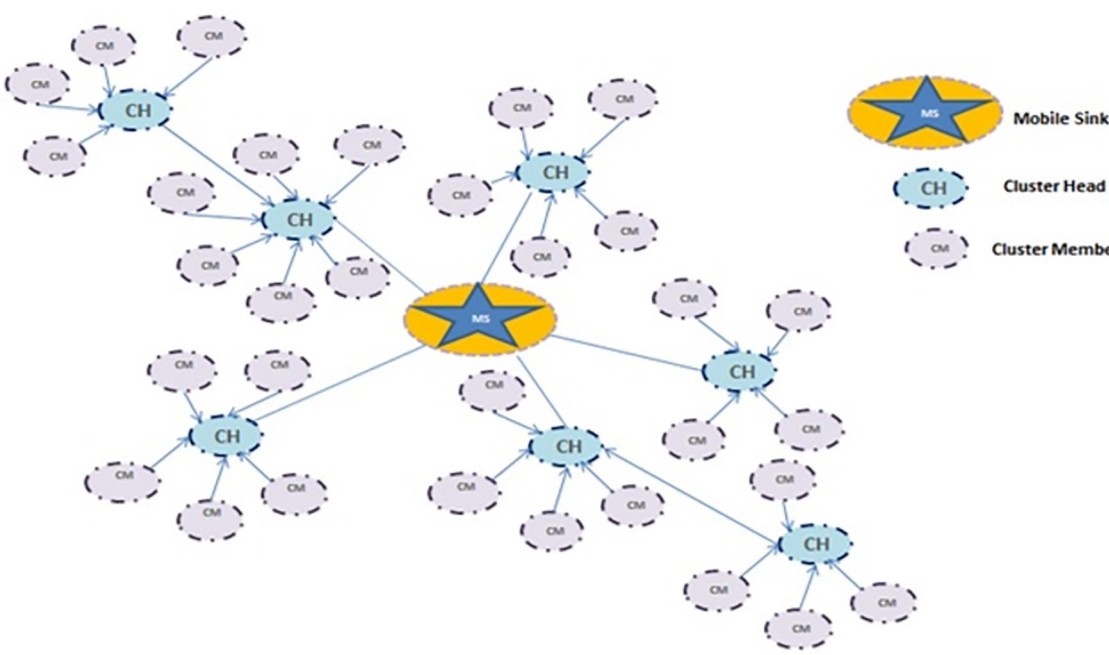

**Fig 4. Platform configuration of the proposed method.**

**Table 2. Simulation parameters [21].**

| Parameter | Value |
|---|---|
| Distributing nodes in the environment | Random |
| The mobility models of Sink | Random |
| Number of sink | 1 |
| Number of nodes | 60 |
| Simulation environment | $100*100*100 \, m^3$ |
| Transmitted type | CBR |
| Radio transmission board | 250m |
| Packet size | 1024 bit |
| Simulation time | 250 Sec |
| Mac layer protocol | IEEE 802.15.4 |
| initial energy level | 200–450 Jul |
| $N_{ed}$ Number of elimination and dispersal events | 10 |
| $N_{re}$ Number of productivity | 100 |
| $N_c$ chemotaxis in slipping or hanging | 10 |
| $d_{attractant}$ Swimming length | 0.1 |
| Probability of elimination and dispersal | 0.8 |
| Searching space dimensions | 3D |
| $h_{repllant}$ | 0.1 |
| $w_{repellant}$ | 10 |
| $w_{attractant}$ | 0.2 |

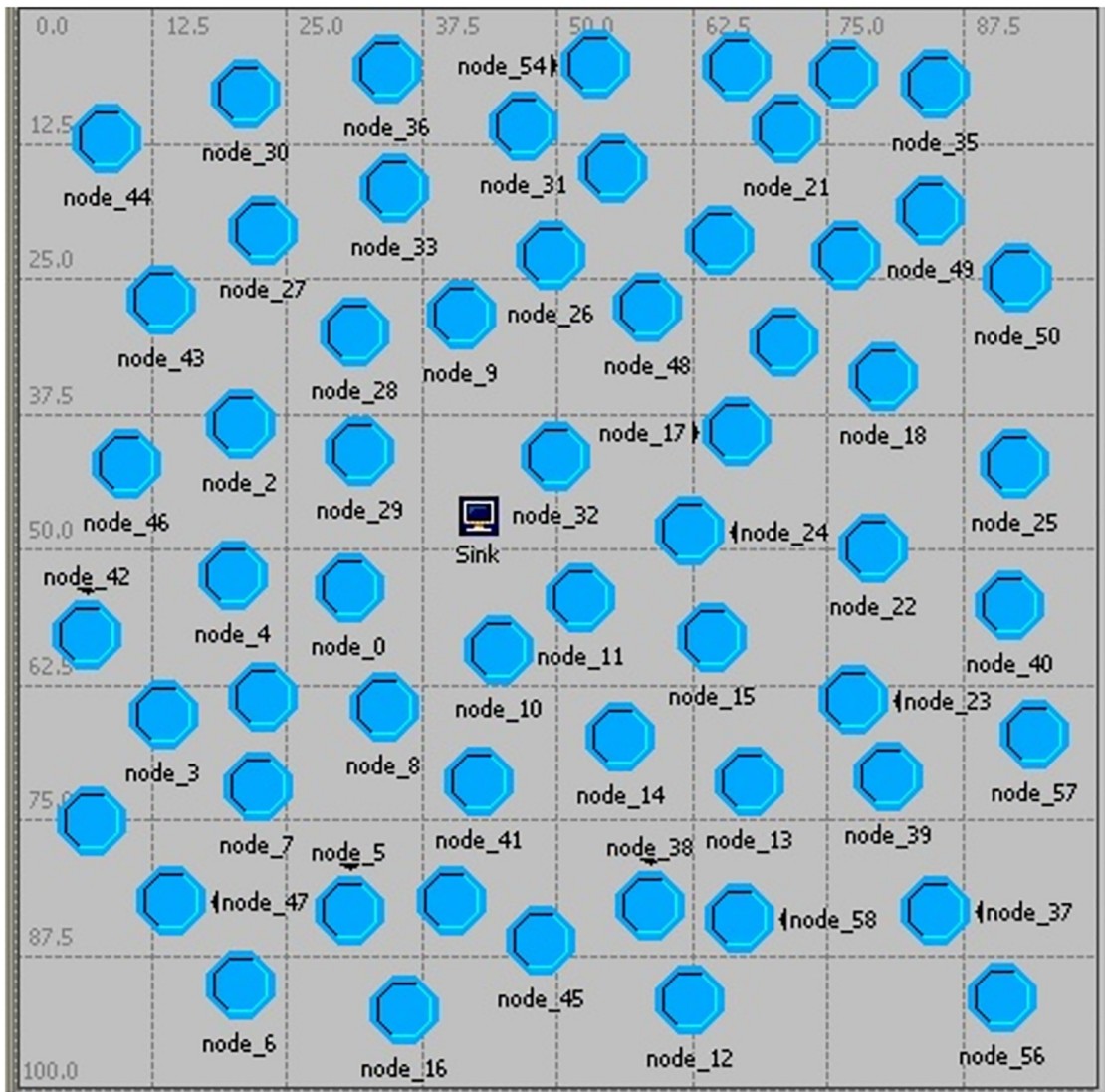

**Fig 5. View of the simulated model network topology.**

- Media access delay: Equals the time between receiving the packet by the MAC layer until it is fully loaded on the wireless media. The reason for checking media access latency is that many multimedia applications have latency constraints and the data is no longer usable after this time.

- Packet error rate: is equal to the average number of bit errors in packets.

- Throughput: Defined as the total number of packets received by the receiver divided by the time between the receipt of the first packet and the last packet. It is equal to dividing the file size at that time, in megabits per second.

- Signal to Noise Rate: The signal-to-noise ratio is a measure of the amount of useful signal versus annoying signal or noise. This number represents the amount of noise imposed on a signal versus the strength of the signal itself. The higher the index, the better and more useful the signal.

- Data packet loss rate: The number of data packets lost due to noise, congestion, or error.

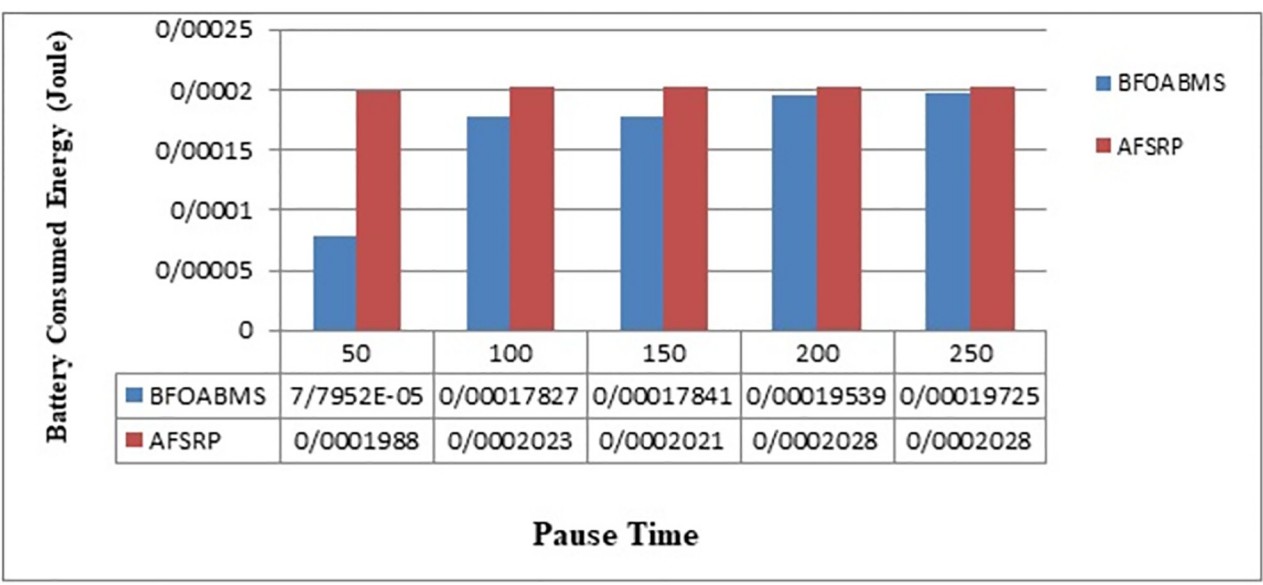

**Fig 6. Average energy consumption.**

## 4 3 Simulation results

Fig 6 compares the average battery power consumption of network nodes for the proposed BFOABMS algorithm scenario and the AFSRP protocol. The vertical axis shows the energy consumption and the horizontal axis shows the simulation time. The energy consumed is equal to the sum of the energy used by the sensor nodes for communication. As can be seen, the AFSRP protocol consumes more energy. In the AFSRP protocol, if the network switches to selecting a cluster head based on the location of the node, a node that is less distant from the sink and the residual energy are not less than the threshold is cluster head, otherwise if a node with this feature is not available. This algorithm does not perform clustering, so the selected cluster heads may have less energy in the next cycles, so select them cluster head, and due to high energy consumption in the cluster heads, these nodes lose their energy quickly, and the network topology Also in this protocol, the node cluster heads send the collected data directly to the sink node. And nodes that are far away from the sink inform the sink if it has a lot of data to move to the denser area. If in the proposed method BFOABMS uses clustering and tracking of the sink by the algorithm, the node bacterium feeding optimization algorithm is used as a clue to send and transfer data that have more energy and less distance to the sink. On the other hand, since member nodes also join the cluster head node due to their distance, so it is not necessary to spend a lot of energy to send data from the node to the cluster head. In the proposed method, the use of clustering and mobile sink causes data to be sent from a short path and thus improves energy consumption.

Fig 7 compares the end-to-end latencies for the proposed BFOABMS method scenarios and the AFSRP protocol scenario. The vertical axis is the end-to-end delay and the horizontal axis is the simulation time. End-to-End delay is the time it takes for a data packet to be transmitted from sender to receiver. As can be seen in the AFSRP protocol in the same initial cycles, the energy of the threaded nodes may decrease and it may not be able to transmit the sensed information to the thread before the sink reaches it, so the delay is long. However, in the proposed

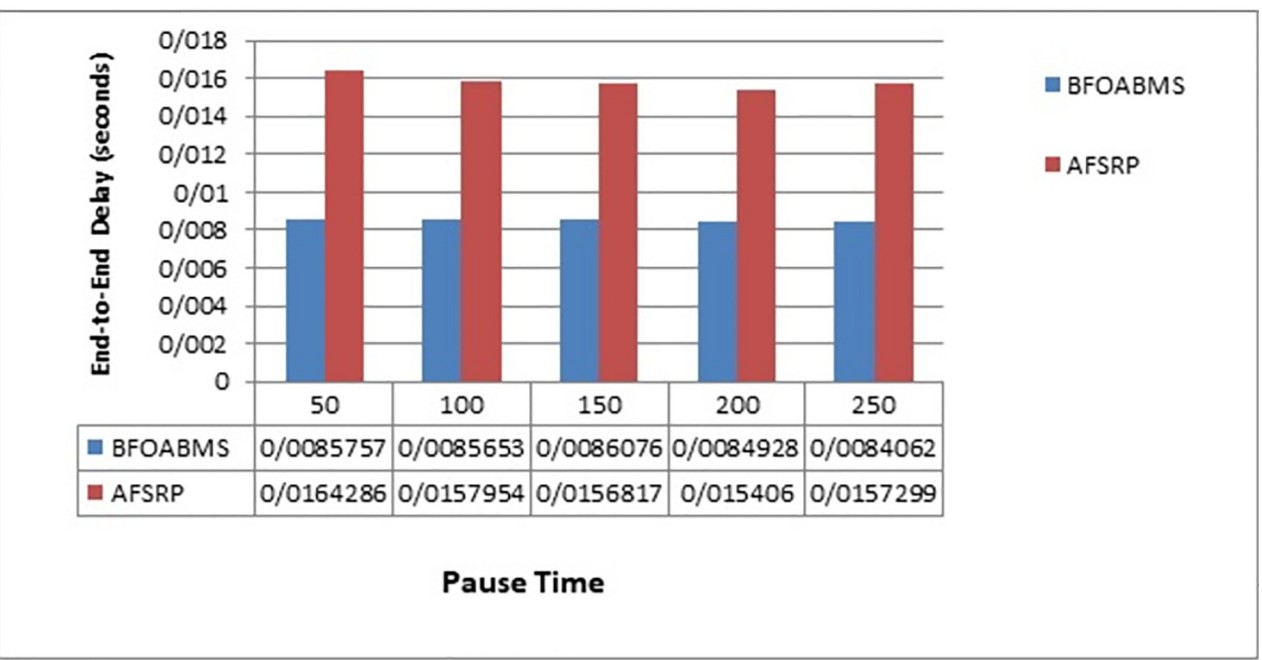

**Fig 7. End-to-end delay.**

BFOABMS method, because in clustering the clusters are selected from the nodes with more energy and less distance to the moving sink and the cluster members also join the cluster in terms of distance, so the end-to-end delay is reduced. Using the mobile sink, which travels between sensor nodes that have connection problems, can improve the connectivity between sensor nodes and base station, and reduce the end to end delay in data transfer.

Fig 8 shows the throughput rates for the two scenarios. The horizontal axis represents the simulation time and the vertical axis the number of data bits delivered at the time or throughput rate. The throughput is equal to the total packets received by the receivers divided by the time between receiving the first packet and the last packet. According to Fig 8 of the AFSRP protocol, compared to the proposed BFOABMS method, the number of packets successfully delivered to the sink is lower than the total packets transmitted by the sensor nodes due to congestion and possible node shutdown. However, in the proposed BFOABMS method, due to the use of a moving sink and sending data through cluster head to the sink with a shorter distance, the throughput rate increases. Therefore, the path found to the sink does not change until the end of the data transfer phase. Therefore, the number of packets delivered to the sink node will be higher in the proposed method. The proposed method can find optimal visiting points and data gathering paths for a mobile sink within clusters and define an optimal clustering and data gathering path, then it can improve the data collection and throughput performance.

Fig 9 shows the signal-to-noise ratio for the proposed BFOABMS method scenarios and the AFSRP protocol scenario. The horizontal axis shows the simulation time and the vertical axis the signal-to-noise ratio. The signal-to-noise ratio is used to indicate the amount of useful signal against the annoying signal or noise. The higher this criterion, the more useful the signal. According to Fig 9, the AFSRP protocol has a lower signal-to-noise ratio than the proposed BFOABMS method because the AFSRP protocol may increase the number of erroneous bits

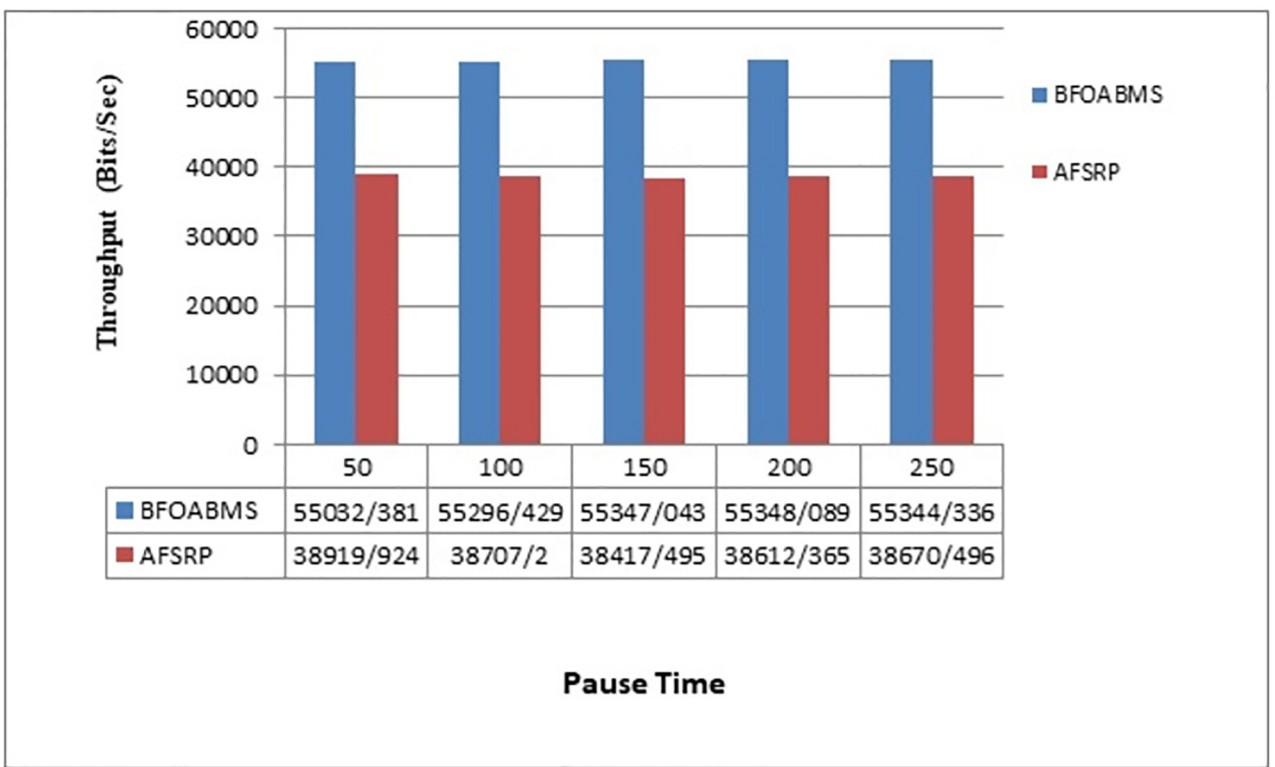

**Fig 8. Throughput rate.**

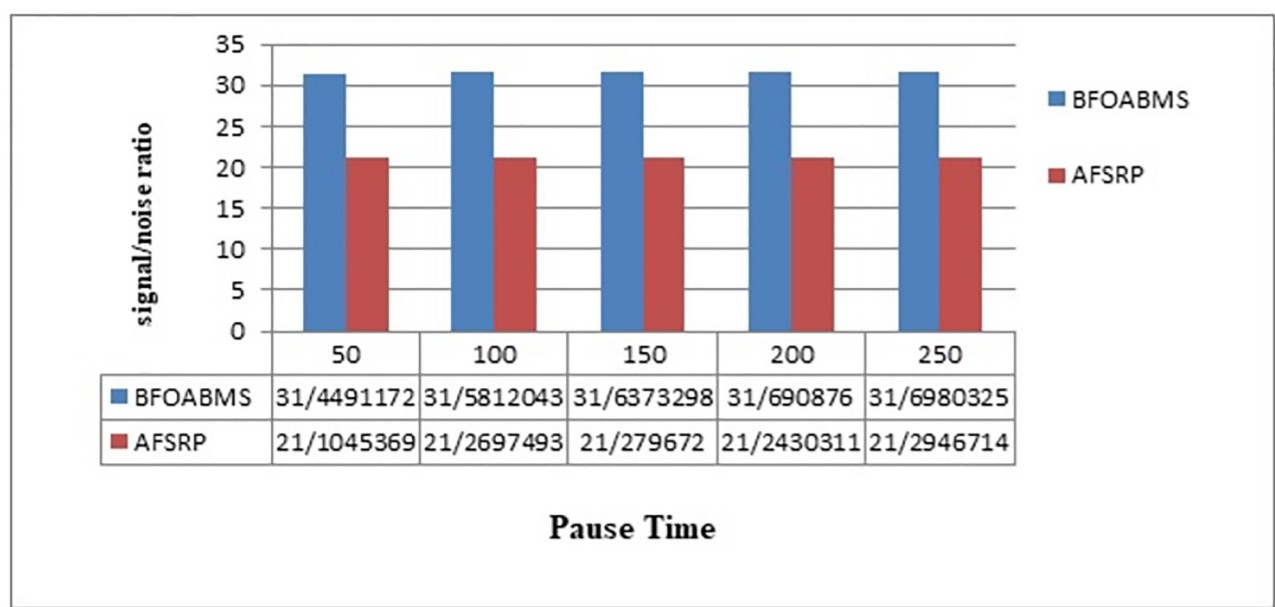

**Fig 9. Signal to noise ratio.**

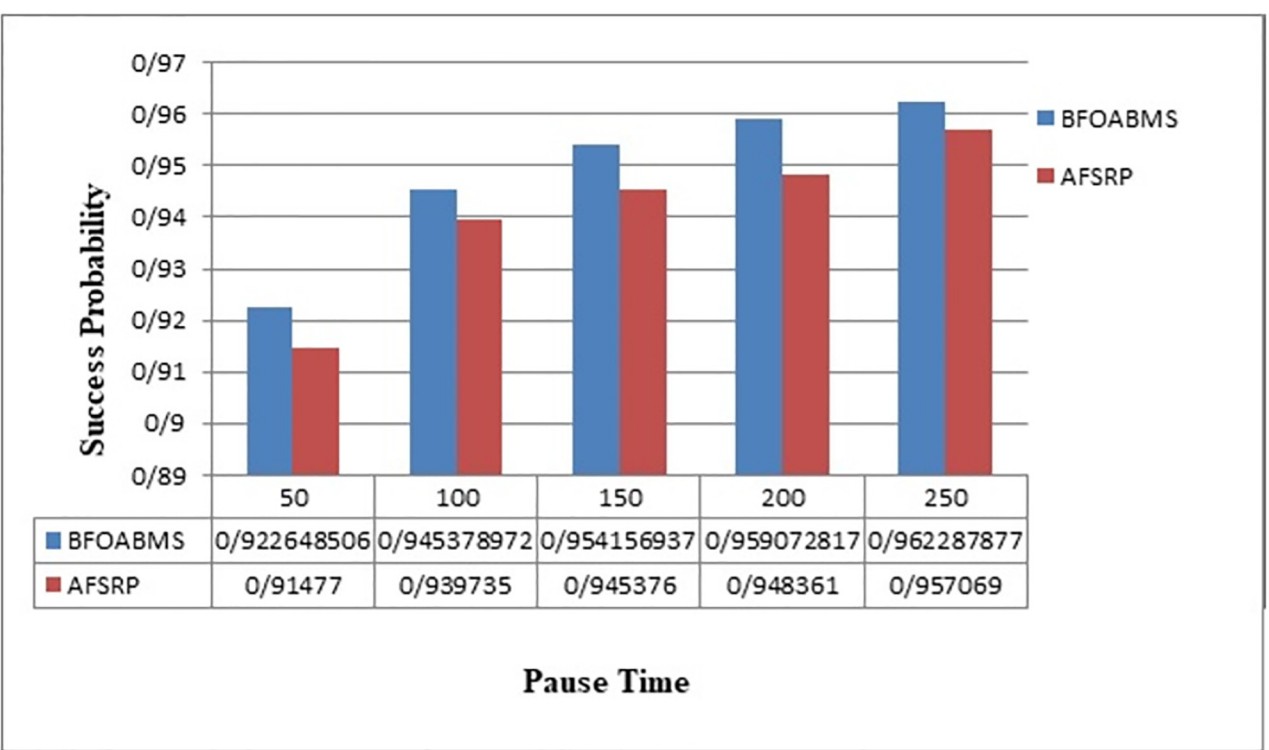

**Fig 10. Delivery rate successfully closes data to the sink.**

when sending and reduce the signal-to-noise ratio. Also, the information signal sent by the AFSRP protocol may be lost due to congestion and disturbance, and the possibility of noise may increase, thus reducing the quality of the transmitted data. In the proposed BFOABMS method, data is sent through the cluster head close to the moving sink, thus reducing the likelihood of disturbance and increasing the signal-to-noise ratio. signal-to-noise ratio is used to determine the quality of network connections, and the higher value indicates that the node has higher signal power. The larger the signal-to-noise ratio is, the higher probability of continuing bonding for a longer time. in the proposed method using the mobile sink and clustering improves network lifetime, signal-to-noise ratio, and channel capacity.

Fig 10 compares the probability of successfully sending data to the sink node for the proposed BFOABMS method and the AFSRP protocol scenario. In a wireless sensor network, it is a simple way to route a single step and send data directly to the base station, which is very costly because nodes that are far from the sink may drain their energy storage faster. Therefore, severely limit the life of the network. According to Fig 10, as seen in the AFSRP protocol, the transmission success rate is low because some network nodes may be shut down due to a hardware error or battery drain and the data transfer to the sink node may not be completed, but in the proposed BFOABMS method Due to the choice of high stability path that contains high energy nodes and on the other hand due to the mobility of the sink, the probability of sending data to the sink increases. The sensor nodes close to the sink have more traffic loads, then they will quickly deplete their energy. using the mobile sink can improve energy consumption for

WSNs. The mobile sink will sojourn at some fixed points to collect raw data from relevant areas then it can increase data transfer of WSNs.

## 5. Conclusion

In this paper, a new algorithm is proposed that clusters sensor nodes in wireless sensor networks using the bacterial foraging optimization algorithm to improve the energy consumption problem. This method identifies several nodes as cluster heads, which leads to the formation of suitable clusters in the network. in clusters, nodes adopt a one-step routing scheme to communicate with the cluster head, which, when receiving data packets from all members of the cluster, transmits the collected data to the mobile sink node in a pre-calculated path. In inter-cluster communication, the neighboring node is selected with the maximum residual energy. The proposed method prevents the transfer of data to the sink over long distances, thus minimizing the network energy consumption. On the other hand, in sensor networks with fixed sink, because the nodes close to the sink due to excessive use of their battery energy to send data, other sensor nodes in the network are more likely than other nodes to drain their battery resources which this leads to topological failure and reduced sensory coverage, thus increasing the data loss. Accordingly, the use of a moving sink can solve this problem. The proposed method and AFSRP [21] in the OPNET simulator simulate and the results show that the proposed method for network features such as improved power consumption, packet loss rate Data, throughput rates, end-to-end latency works better.

## Supporting information

**S1 Dataset.**
(RAR)

## Author Contributions

**Conceptualization:** Shayesteh Tabatabaei.

**Formal analysis:** Shayesteh Tabatabaei.

**Funding acquisition:** Shayesteh Tabatabaei.

**Investigation:** Shayesteh Tabatabaei.

**Methodology:** Shayesteh Tabatabaei.

**Project administration:** Shayesteh Tabatabaei.

**Resources:** Shayesteh Tabatabaei.

**Software:** Shayesteh Tabatabaei.

**Supervision:** Shayesteh Tabatabaei.

**Validation:** Shayesteh Tabatabaei.

**Visualization:** Shayesteh Tabatabaei.

**Writing – original draft:** Shayesteh Tabatabaei.

**Writing – review & editing:** Shayesteh Tabatabaei.

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
