## [Decision Letter · Decision Letter 0]

21 Jul 2021

PONE-D-21-15940

Provide energy-aware routing protocol in wireless sensor networks using bacterial foraging optimization algorithm and mobile sink

PLOS ONE

Dear Dr. Tabatabaei,

Thank you for submitting your manuscript to PLOS ONE. After careful consideration, we feel that it has merit but does not fully meet PLOS ONE’s publication criteria as it currently stands. Therefore, we invite you to submit a revised version of the manuscript that addresses the points raised during the review process.

A rebuttal letter that responds to each point raised by the academic editor and reviewer(s). You should upload this letter as a separate file labeled 'Response to Reviewers'.A marked-up copy of your manuscript that highlights changes made to the original version. You should upload this as a separate file labeled 'Revised Manuscript with Track Changes'.An unmarked version of your revised paper without tracked changes. You should upload this as a separate file labeled 'Manuscript'

We look forward to receiving your revised manuscript.

Kind regards,

Ping He, Ph.D.

Academic Editor

PLOS ONE

Journal Requirements:

4. We note that you have stated that you will provide repository information for your data at acceptance. Should your manuscript be accepted for publication, we will hold it until you provide the relevant accession numbers or DOIs necessary to access your data. If you wish to make changes to your Data Availability statement, please describe these changes in your cover letter and we will update your Data Availability statement to reflect the information you provide

5. Your abstract cannot contain citations. Please only include citations in the body text of the manuscript, and ensure that they remain in ascending numerical order on first mention.

Additional Editor Comments (if provided):

The quality of this manuscript is terrible. Seriously hindered my judgment of the manuscript work. Please read the comments carefully, polish your manuscript in an all-round way, and then submit it again.

Reviewers' comments:

Reviewer's Responses to Questions

**Comments to the Author**

1. Is the manuscript technically sound, and do the data support the conclusions?

Reviewer #1: Yes

Reviewer #2: No

Reviewer #3: Partly

Reviewer #4: Partly

Reviewer #5: Yes

Reviewer #6: No

Reviewer #7: Partly

Reviewer #8: Yes

Reviewer #9: Partly

Reviewer #10: Yes

2. Has the statistical analysis been performed appropriately and rigorously? 

Reviewer #1: Yes

Reviewer #2: No

Reviewer #3: No

Reviewer #4: I Don't Know

Reviewer #5: Yes

Reviewer #6: N/A

Reviewer #7: N/A

Reviewer #8: N/A

Reviewer #9: N/A

Reviewer #10: Yes

3. Have the authors made all data underlying the findings in their manuscript fully available?

Reviewer #1: Yes

Reviewer #2: Yes

Reviewer #3: Yes

Reviewer #4: Yes

Reviewer #5: No

Reviewer #6: No

Reviewer #7: No

Reviewer #8: Yes

Reviewer #9: Yes

Reviewer #10: Yes

4. Is the manuscript presented in an intelligible fashion and written in standard English?

Reviewer #1: Yes

Reviewer #2: No

Reviewer #3: Yes

Reviewer #4: No

Reviewer #5: Yes

Reviewer #6: No

Reviewer #7: Yes

Reviewer #8: Yes

Reviewer #9: No

Reviewer #10: Yes

5. Review Comments to the Author

Reviewer #1: Improve on the following:

Problem statement

Theoretical background/framework

Use more studies in the literature review

PARTICULARLY, IMPLICATIONS ARE NOT ADEQUATELY DISCUSSED. THIS IS THE WEAKEST POINT.

Reviewer #2: Suggestions for improvement:

1) The Introduction and related works need to be revised and should emphasize the challenges and corresponding techniques.

2) Proposed implementation method lacks detailed information and proper explanation.

3) Fig. 3 needs to be written as an algorithm and explained step by step.

4) Results section can be further improved. Authors are recommended to evaluate their proposed method with the latest existing and add more performance evaluation metrics too.

Reviewer #3: Comments and Suggestions for Authors

The article describes a new algorithm is proposed that clusters sensor nodes in wireless sensor

networks using the bacterial foraging optimization algorithm to improve the energy consumption

problem.

Major comments:

It seems there is much room for improvement before the publication. I list my major comments on

the manuscript in the attached file.

Reviewer #4: The ID of the manuscript : PONE-21-15940

The title of the manuscript : Provide energy-aware routing protocol in wireless sensor networks using bacterial foraging optimization algorithm and mobile sink

Manuscript Summary :

In this work, a wireless sensor network routing algorithm is proposed. The proposed algorithm utilizes the ultra-innovative algorithm for bacterial Foraging and mobile sink, which leads to energy efficiency. In this work, two criteria are used to determine the number of sensor nodes, which are: the amount of energy on the battery surface and the distance to the sink as a head, which leads to the formation of regular clusters in the network. Within the network, nodes adopt a multi-step routing scheme to communicate with the sink. On WSNs with fixed sinks, nodes close to the sink are more likely than other nodes to share multi-step paths and focus data toward the sink and they drain their energy grid. Shutting down nodes leads to topology failure and disrupts the reporting of sensor data. To deal with this problem, the use of mobile sinks is used to be able to balance the load and help to consume uniform energy throughout the network. The proposed protocol operates in a distributed manner and can reduce energy consumption.

The manuscript is interesting; however, the following comment need to be addressed :

1 – Check the English grammar mistakes and correct them .

2 – The introduction section is a single paragraph, which makes it difficult to read. Try to divide it into several paragraphs .

3 – At the end of the introduction section, the contribution list should be included and stated clearly .

4 – In the related work section, recent algorithm need to be included such as:

[I] A. S. Al-Zubaidi, B. M. Mahmmod, S. H. Abdulhussain, and D. Al-Jumaeily, “Re-evaluation of the stable improved LEACH routing protocol for wireless sensor network,” in Proceedings of the International Conference on Information and Communication Technology - ICICT ’19, 2019, pp. 96–101.

[II] Salem, Amer O. Abu, and Noor Shudifat. "Enhanced LEACH protocol for increasing a lifetime of WSNs." Personal and Ubiquitous Computing 23.5 (2019): 901-907.

5 – The paragraphs need to be divided into several paragraph for more readability .

6 – A table need to be included at the end of the related work section. The table summarizes the algorithm in the section as well as includes the highlights and limitations of each algorithm .

7 – symbol should be included in math mode not in regular text mode .

8 – soe equation are bold and not aligned correctly . please correct them .

9 – each variable in the manuscript need to be defined clearly .

10 – Try not to use “_”, the subscript can be used.

11 – Figures 1 need to be enlarged. In addition, the text is not clear inside the blue color, try to select clear colors .

12 – Figure 3 should be replaced by algorithm style. In addition, the algorithm is not clear and not given in a professional manner .

13 – more discussion about the results should be included .

14 – missing state of the art algorithm in the comparison . Try to include some .

Reviewer #5: May limit the length of the abstract.

'Related work' chapter is vast.

BFO is poor convergence behaviour - It is required to describe with respect your problem. What is the remedy you have taken etc..

What happened if you make it your algorithm as self adaptive. Though you are not implemented it, just describe what happened if it is incorporated. (Since it is crucial parameters in SFO)

Pictorial view of your problem may be included. (this will attract more viewers)

elimination is non-uniform or uniform?

How the test run is performed?

statistical tests need to be performed to determine the stability of the algorithm and identify the best value of the parameter.

Is it this is metaheuristic?

Your algo. is seems to be Multiobjective .....energy, distance etc - justify.

What is the strategy of E.coli in your algo.

BFO has advantages, such as parallel distributed processing, insensitivity to initial value, and global optimization. How these are helping your algorithm.

Reviewer #6: This paper suggests the use of heuristic optimization to select routes of data from sensor nodes to a sink node in a WSN with the goal of improving the energy efficiency of routing.

Major concerns:

The write up and presentation of the paper are poor; sometimes very poor to the extent that the description is incomprehensible.

The validation of the work is not sound. It is based entirely on comparing simulation results with a single work that was published just this year by one of the authors ignoring all other established works and results. It is also based on a single scenario and topology with no justification of parameter selection.

There are also several questionable points about the simulation setup and evaluation criteria. For example, how can sensor nodes transfer MB sized files and why is there a test for multimedia content as found in multimedia applications (when measuring media access delay). What type of sensor nodes are we talking about?

The novelty of the work is not clear at all. To the contrary, the same ideas have been presented many times even by at least one of the authors. I have noticed a common theme in the publications by the team where a series of the same ideas are presented using various metaheuristic algorithms and always benchmarking with a single work using the same simulation setup. This adds nothing more than noise to the body of knowledge.

There is no clear justification or analysis for the use of yet another bio-inspired optimization algorithm to select the paths from sensor nodes to sink nodes or to form clusters. The modeling part is not clear and incomprehensible: the cost function and parameter selection are not well-explained and not clearly justified. Parameters are given values that seem (off-the-shelf) with no explanation. In fact, the core optimization algorithm that is described as the proposed method is listed in figure 3 as a snapshot from a printed book!

The related work section has almost no value mainly because of the poor write up and lack of analysis or at least proper description that can be understood by the reader without referring to the original work. All the listed related work and their results have been ignored later on except one paper co-authored by one of the authors and used as the sole benchmark to validate and evaluate the simulation.

Reviewer #7: 1. The authors should select better algorithms in recent paper for the comparison of results.

2. Include the references from reputed journals for the past three years

3. The author should do the simulation with various number of nodes and areas.

Reviewer #8: The manuscript proposes an algorithm based on bacterial foraging for energy efficiency in wireless sensor network routing.

The method proposed is explained in detail and the simulation results are given. The results are compared to AFSRP method and they are superior.

There are few minor problems with the manuscript: Equations 3 and 4 have to be reorganized, they are misplaced; I guess Figure 2 has missing parts and it looks as Table rather than Figure: Figure 3 could be given as Table; The numbering of sections 4.3 and 5 are wrong. Section 2 is too long it can be shortened and a table could be given to ease the follow up of the literature.

Reviewer #9: Comments to the Author

In this paper Authors have discussed about a new algorithm to improve the energy

consumption problem by forming suitable clusters in the network. They use BFOA in their work. I am presenting my critical view of the work with the hope that the authors will appreciate the angle from which I am viewing the work and I am hoping that my view would help them in strengthening their work further.

1. Abstract needs to be compact and clearly describe the work.

2. In the introduction part the WSN must be explained using suitable pictures. It is also required to describe about the cluster design by using pictures.

3. The section-2 (Related work) must be re-written. In that section authors must describe only the published previous work without comparing their work. They can use a comparison table to compare their work with these published articles in a separate place. It will increase the readability of their work.

4. Some the abbreviation used without their full name anywhere in the paper. E.g. DCRRP.

5. Reference [8] is not cited in the text.

6. Figure -3 in the current form is not acceptable.

7. Authors may use the same font throughout their paper. e.g reference [18] is written in a different font.

Reviewer #10: In the manuscript the algorithm for routing in the wireless sensor network is proposed using the ultra- innovative algorithm for bacterial Foraging and mobile sink, which leads to energy efficiency, the number of sensor nodes is determined according to two criteria: the amount of energy on the battery surface and the distance to the sink.

The paper is well written. Here are few observations.

Include a separate subsection to highlight how your work differs from the available literature.

The block diagram has to be explained.

The simulation scenario such placing of nodes should be explained.

Practical applicability of the proposed optimization set up should be explored.

How do you compare with the state of art optimization methodologies.

Are the parameters given in Table 1 chosen randomly or experimentally, if so how does the choice effects the throughput performance.

The pseudocode is poorly represented.

The pseudocode should be explained, since there is lack of cordination between the methodology section and the pseudocode.

Proper citations to the equations should be given.

The paper must be corrected for typo and grammatical errors.

6. PLOS authors have the option to publish the peer review history of their article (what does this mean?). If published, this will include your full peer review and any attached files.

Reviewer #1: No

Reviewer #2: No

Reviewer #3: **Yes: **HOSAM ALRAHHAL

Reviewer #4: No

Reviewer #5: No

Reviewer #6: **Yes: **Ahmed Al-Haiqi

Reviewer #7: **Yes: **Kalpana Murugan

Reviewer #8: No

Reviewer #9: **Yes: **Om Prakash Acharya

Reviewer #10: No

---

## [Author Response · Author response to Decision Letter 0]

10 Sep 2021

September 02, 2021

Dear Dr. Ping He,

Re your email, I am writing to submit the revised version of the manuscript entitled “Provide energy-aware routing protocol in wireless sensor networks using bacterial foraging optimization algorithm and mobile sink ” (ID: PONE-D-21-15940 ).

I would like to take this opportunity to express my appreciation for the invaluable comments and suggestions which you and the reviewer kindly provided. I tried my best to revise the manuscript in accordance with the comments. Detailed corrections are listed below point by point. Also, I have highlighted the changes I made in the manuscript. I hope the revised form of the manuscript meets the expectation of the reviewers and is suitable for publication in “PLOS ONE.”

Thank you again for your consideration.

I am looking forward to reading from you.

Sincerely yours,

Dr. Sh.Tabatabaei 

………………………………………………………………………………………………………

Responses to Comments

Reviewer

# Comment Response

Reviewer #1

1 The innovation and the motivation behind this work are not clearly highlighted. Please work on this and prove to us why this work is valuable. And the novelty of the proposed model is questionable. Correction was made and added innovation and the motivation this work in introduction.

2 The contributions and novelty of this paper are not listed in the introduction section Correction was made.

3 On page 8, the claim (Wireless sensor networks (WSNs) include small sensor nodes with battery and processing power and limited memory units. There are two important features of WSNs, one is infrastructure-free and the other is self-organizing.) need to be supported by references. Correction was made and the reference is added in the text.

4 The authors used a number of abbreviations without clarification the original words of these abbreviations, for example, but not limited to “DCRRP”, “OPNET”, “SFLA”, and “FIS” The original words of abbreviations“DCRRP”, “OPNET”, “SFLA”, and “FIS” are added.

5 On page 9, the claim (The advantage of this method is the use of an optimized aggressive weed algorithm that can adapt and be random and can quickly find the global optimal, as well as the use of a fuzzy inference model that has high accuracy. The disadvantage of this 

method is the high processing cost due to the use of fuzzy logic with an aggressive weed algorithm.) need to be supported by references. Correction was made and the reference is added in the text.

6 The reference [8] is missed in the text. The reference [8] is added in the text.

7 OPNET 11.5 Simulator and BIFOA need references. Correction was made and the reference [24] is added for OPNET 11.5 also the reference [22] is added for BIFOA. 

8 In general, The values of parameters (for example, s, p, Wattractant, dattractant , wrepllant, drepllant, the number of sensor nodes is 60, and the values in table 1 ) were selected arbitrarily. The authors provide no guidance on how and why to select these values, no references supported that. The simulation result of the proposed protocol is compared with the AFSRP (Artificial Fish Swarm Routing Protocol) then all parameter is similar with the reference [21]. Also, the reference [21] is added for the parameter in table 1.

9 Correct the writing ith, jth, and kth. Also, correct hrepllant -> hrepllant and Wreplant -> Wrepllant. Correction was made.

10 Equations1 and 2 need references. Correction was made and Equations 1 and 2 are referenced with [22].

11 In this sentence (In this regard, Wrepllant, Wattractant, Wattractant, dattractant are different coefficients that must be selected correctly.) Wattractant is repeated, maybe it must change to hrepllant . Correction was made and Wattractant changed to hrepllant . 

12 On page 14, where is Equation 3-2? Correction was made and Equation 3-2 changed to Equation 2.

13 On page 14, the claim (However, because the Wattractant value is always considered less than the Wrepllant.) why? And needs to be supported by references. Correction was made and the reference is added.

14 In Figure 1, the direction of the arrow in the loop is needed. The direction of the arrow in the loop is added in Figure 1. 

15 Adjust Equations 3 and 4. Also, adjust the format of Table 1. Correction was made.

16 On page 16, step 2, ie -> i.e. Correction was made.

17 On page 16, step 3, selects fifty percent, why? It needs a reference Because in the bacterial foraging optimization algorithm selects fifty percent of the bacteria, then we selects fifty percent in step 3, reference is added.

18 On page 17, step 4, ped -> Ped. Correction was made.

19 In step 6, if there is a malicious node, and participated in the process, what the effect on the proposed method, and how it treats this issue? With consideration, that the sink has a table content of IDs of all sensor nodes and the sink sends a notification message of being cluster head to cluster head sensors, then if malicious node want to participate in the process, sink detected it, and sends IDs of malicious node for all selected cluster head to not accept join request from this malicious node.

20 On page 17, (If the residual energy of the sprigs is more than 70% of their total initial energy,). Why you selected this value? This claim needs to support by references. Correction was made and the reference [9] is added. This sentence" If the residual energy of the sprigs is more than 70% of their total initial energy "is in reference [9].

21 On page 21, 3-4 Simulation results -> 4-3 Simulation results. Correction was made.

22 On page 21, simulation results subsection, on line 8, there is a comma (,) it must remove it. Correction was made and comma (,) removed.

23 What are the mobility and energy consumption models used in this paper? The mobility model is random, it is added to table 1 and energy consumption equation number 6 is added in the text.

24 Platform configuration and operating system are used to test run are also not stated very well. Platform configuration is show in fig 4 and we use Contiki operating system that it uses standard C and follows an event-driven programming model that added in the text.

25 In Figure 5, the values of the two methods are approximately similar, is that mean the proposed method not efficient when the value of pause time is high? The proposed method is efficient for all constant time or all simulation, also it is efficient when the value of time is high in figure 5 (energy consumption) shows only one pause time may the proposed method and the AFSRP method is maybe similar.

26 In Figures 5, and 6, the horizontal axis shows the simulation time or the pause time? In Figures 5, and 6, the horizontal axis shows the pause time.

27 In Figure 6, the values of end-to-end delay are approximately similar for all pause time values, is that mean end-to-end delay not affected by pause time? If yes, why you study it? Attention in Figure 6, for the proposed method the values of end-to-end delay for all pause time and all simulation time value is efficient and better than the AFSRP method.

28 The results in Figures 6 and 7 are questionable because when the value of pause time is small that means the mobility of the nodes in the network is high, so the probability to break the links between nodes will increase. As a result, the throughput will decrease and the delay will increase in proposed method all nodes is fixed and no mobility for the nodes only sink is mobiled in the network, other wise the sink will make a decision based on the residual energy of the sprigs in its table, If the residual energy of the sprigs is less than 70%, the sink moves to a denser area then the cluthe probability to break the links between nodes will decrease. As a result, the throughput will increase. also with the moves of sink toward denser area the cluster heads can send data with the nearest way to sink then decrease delay.

………………………………………………………………………………………………………

---

## [Decision Letter · Decision Letter 1]

29 Nov 2021

PONE-D-21-15940R1Provide energy-aware routing protocol in wireless sensor networks using bacterial foraging optimization algorithm and mobile sinkPLOS ONE

Dear Dr. Tabatabaei,

Thank you for submitting your manuscript to PLOS ONE. After careful consideration, we feel that it has merit but does not fully meet PLOS ONE’s publication criteria as it currently stands. Therefore, we invite you to submit a revised version of the manuscript that addresses the points raised during the review process.

We look forward to receiving your revised manuscript.

Kind regards,

Ping He, Ph.D.

Academic Editor

PLOS ONE

Additional Editor Comments:

The authors did not address any of the comments raised in previous round; thus, there are remains unsolved. Give the author one last chance to revise. If the author cannot complete the revision as required, the manuscript will be rejected for publication. This will be the final decision.

Reviewers' comments:

Reviewer's Responses to Questions

**Comments to the Author**

1. If the authors have adequately addressed your comments raised in a previous round of review and you feel that this manuscript is now acceptable for publication, you may indicate that here to bypass the “Comments to the Author” section, enter your conflict of interest statement in the “Confidential to Editor” section, and submit your "Accept" recommendation.

Reviewer #1: All comments have been addressed

Reviewer #3: (No Response)

Reviewer #4: (No Response)

Reviewer #5: All comments have been addressed

Reviewer #7: All comments have been addressed

2. Is the manuscript technically sound, and do the data support the conclusions?

Reviewer #1: Partly

Reviewer #3: Partly

Reviewer #4: Partly

Reviewer #5: Partly

Reviewer #7: Yes

3. Has the statistical analysis been performed appropriately and rigorously? 

Reviewer #1: (No Response)

Reviewer #3: N/A

Reviewer #4: Yes

Reviewer #5: N/A

Reviewer #7: Yes

4. Have the authors made all data underlying the findings in their manuscript fully available?

Reviewer #1: Yes

Reviewer #3: Yes

Reviewer #4: No

Reviewer #5: Yes

Reviewer #7: Yes

5. Is the manuscript presented in an intelligible fashion and written in standard English?

Reviewer #1: Yes

Reviewer #3: Yes

Reviewer #4: No

Reviewer #5: Yes

Reviewer #7: Yes

6. Review Comments to the Author

Reviewer #1: The manuscript has improved and my comments are addressed particularly in terms of novelty, and purpose statement.

Reviewer #3: Minor comments:

1- The contributions and novelty of this paper are not listed in the introduction section.

2- Review the paper to be sure the font is the same for all the paper.

3- Adjust Equations1 and 2, also put their references in a suitable place. Also, for Equation 6.

4- Adjust Equations 3 and 4.

Reviewer #4: The ID of the manuscript : PONE-21-15940R1

The title of the manuscript : Provide energy-aware routing protocol in wireless sensor networks using bacterial foraging optimization algorithm and mobile sink

Manuscript Summary :

In this work, a wireless sensor network routing algorithm is proposed. The proposed algorithm utilizes the ultra-innovative algorithm for bacterial Foraging and mobile sink, which leads to energy efficiency. In this work, two criteria are used to determine the number of sensor nodes, which are: the amount of energy on the battery surface and the distance to the sink as a head, which leads to the formation of regular clusters in the network. Within the network, nodes adopt a multi-step routing scheme to communicate with the sink. On WSNs with fixed sinks, nodes close to the sink are more likely than other nodes to share multi-step paths and focus data toward the sink and they drain their energy grid. Shutting down nodes leads to topology failure and disrupts the reporting of sensor data. To deal with this problem, the use of mobile sinks is used to be able to balance the load and help to consume uniform energy throughout the network. The proposed protocol operates in a distributed manner and can reduce energy consumption.

The author misses to address the comments raised from the previous round.

The manuscript is interesting; however, the following comment need to be addressed :

1 – Check the English grammar mistakes and correct them .

2 – The introduction section is a single paragraph, which makes it difficult to read. Try to divide it into several paragraphs .

3 – At the end of the introduction section, the contribution list should be included and stated clearly .

4 – In the related work section, recent algorithm need to be included such as:

[I] A. S. Al-Zubaidi, et al. “Re-evaluation of the stable improved LEACH routing protocol for wireless sensor network,” in Proceedings of the International Conference on Information and Communication Technology - ICICT ’19, 2019, pp. 96–101.

[II] Salem, Amer O. Abu, and Noor Shudifat. "Enhanced LEACH protocol for increasing a lifetime of WSNs." Personal and Ubiquitous Computing 23.5 (2019): 901-907.

5 – The paragraphs need to be divided into several paragraph for more readability .

6 – A table need to be included at the end of the related work section. The table summarizes the algorithm in the section as well as includes the highlights and limitations of each algorithm .

7 – symbol should be included in math mode not in regular text mode .

8 – some equation are bold and not aligned correctly . please correct them .

9 – each variable in the manuscript need to be defined clearly .

10 – Try not to use “_”, the subscript can be used.

11 – Figures 1 need to be enlarged. In addition, the text is not clear inside the blue color, try to select clear colors .

12 – Figure 3 should be replaced by algorithm style. In addition, the algorithm is not clear and not given in a professional manner .

13 – more discussion about the results should be included .

14 – missing state of the art algorithm in the comparison . Try to include some .

Reviewer #5: Too lengthy abstract. too many introduction. Check with Journal policy.

Check the literature After 2019.

Introduction section may split two or three paragraphs.

Introduction must summarize the existing method's limitations. and then your highlighted contributions.

Figure 2 is similar to table?

Equations should be properly edited. (i.e) 3, 4 etc

Figure 3 should be typed version. Not In the present format.

Since the energy is major focus of your work

- Analyse should include energy vs no of nodes /Analyse should include energy vs no of sensors

- Analyse should include energy vs distance

- Analyse should include energy vs simulation time

Illustration of your work or general block diagram of your approach must be given

Reviewer #7: 1. The innovation and the motivation behind this work are clearly highlighted.

2. The manuscript technically sound piece of scientific research with data that supports the conclusions.

3. Methodology was clearly mentioned with appropriate technique.

4. The author changed the manuscript in response to the reviewer's suggestions.

5. The flow of the concept in an understandable manner.

6. Hence, the paper can be accepted.

7. PLOS authors have the option to publish the peer review history of their article (what does this mean?). If published, this will include your full peer review and any attached files.

Reviewer #1: **Yes: **Mansour Amini ( Universiti Sains Malaysia)

Reviewer #3: **Yes: **HOSAM ALRAHHAL

Reviewer #4: No

Reviewer #5: No

Reviewer #7: **Yes: **Kalpana Murugan

---

## [Author Response · Author response to Decision Letter 1]

23 Dec 2021

Dear Dr. Ping He,

Re your email, I am writing to submit the revised version of the manuscript entitled “Provide energy-aware routing protocol in wireless sensor networks using bacterial foraging optimization algorithm and mobile sink ” (ID: PONE-D-21-15940R1 ).

I would like to take this opportunity to express my appreciation for the invaluable comments and suggestions which you and the reviewer kindly provided. I tried my best to revise the manuscript in accordance with the comments. Detailed corrections are listed below point by point. Also, I have highlighted the changes I made in the manuscript. I hope the revised form of the manuscript meets the expectation of the reviewers and is suitable for publication in “PLOS ONE.”

Thank you again for your consideration.

I am looking forward to reading from you.

Sincerely yours,

Dr. Sh.Tabatabaei

---

## [Decision Letter · Decision Letter 2]

24 Feb 2022

Provide energy-aware routing protocol in wireless sensor networks using bacterial foraging optimization algorithm and mobile sink

PONE-D-21-15940R2

Dear Dr. Tabatabaei,

We’re pleased to inform you that your manuscript has been judged scientifically suitable for publication and will be formally accepted for publication once it meets all outstanding technical requirements.

Kind regards,

Chakchai So-In, Ph.D.

Academic Editor

PLOS ONE

Additional Editor Comments (optional):

Reviewers' comments:

Reviewer's Responses to Questions

**Comments to the Author**

1. If the authors have adequately addressed your comments raised in a previous round of review and you feel that this manuscript is now acceptable for publication, you may indicate that here to bypass the “Comments to the Author” section, enter your conflict of interest statement in the “Confidential to Editor” section, and submit your "Accept" recommendation.

Reviewer #1: All comments have been addressed

Reviewer #3: All comments have been addressed

Reviewer #4: All comments have been addressed

Reviewer #5: All comments have been addressed

2. Is the manuscript technically sound, and do the data support the conclusions?

Reviewer #1: Partly

Reviewer #3: Yes

Reviewer #4: Yes

Reviewer #5: Yes

3. Has the statistical analysis been performed appropriately and rigorously? 

Reviewer #1: Yes

Reviewer #3: N/A

Reviewer #4: Yes

Reviewer #5: Yes

4. Have the authors made all data underlying the findings in their manuscript fully available?

Reviewer #1: Yes

Reviewer #3: Yes

Reviewer #4: Yes

Reviewer #5: Yes

5. Is the manuscript presented in an intelligible fashion and written in standard English?

Reviewer #1: Yes

Reviewer #3: Yes

Reviewer #4: Yes

Reviewer #5: Yes

6. Review Comments to the Author

Reviewer #1: The title needs to be revised, especially the word "provide" at the beginning of the title. Most of my previous comments have been addressed by the author.

Reviewer #3: Comments to the Author

1)I suggest thinking -optionally- about the title as "Energy-aware routing protocol in wireless sensor networks using bacterial foraging optimization algorithm and mobile sink ".

2) Equations 3, 4 are not aligned correctly.

Reviewer #4: The ID of the manuscript : PONE-21-15940 – R2

The title of the manuscript : Provide energy-aware routing protocol in wireless sensor networks using bacterial foraging optimization algorithm and mobile sink

Manuscript Summary :

In this work, a wireless sensor network routing algorithm is proposed. The proposed algorithm utilizes the ultra-innovative algorithm for bacterial Foraging and mobile sink, which leads to energy efficiency. In this work, two criteria are used to determine the number of sensor nodes, which are: the amount of energy on the battery surface and the distance to the sink as a head, which leads to the formation of regular clusters in the network. Within the network, nodes adopt a multi-step routing scheme to communicate with the sink. On WSNs with fixed sinks, nodes close to the sink are more likely than other nodes to share multi-step paths and focus data toward the sink and they drain their energy grid. Shutting down nodes leads to topology failure and disrupts the reporting of sensor data. To deal with this problem, the use of mobile sinks is used to be able to balance the load and help to consume uniform energy throughout the network. The proposed protocol operates in a distributed manner and can reduce energy consumption.

The authors have addressed all the raised comments. The manuscript can be accepted .

- - - - - - - - - - - - - - - - - - - - - - - - - - - - - - - - - - - - - - - - - - - - - - - - - - - - - - - - - - - - - - - - - - - - - - - - - - - - - - - - - - - - - - - - - - - - - - - - - - - - - - - - - - - - - - - - - - - - - - - - - - - - - - - - - - - - - - - - - - - - - - - - - - - - - - - - - - - - - - - - - - - - - - - - - - - - - - - -

Reviewer #5: Contributions can be highlighted.

Differences between the suggested method and existing method can be tabulated.

7. PLOS authors have the option to publish the peer review history of their article (what does this mean?). If published, this will include your full peer review and any attached files.

Reviewer #1: No

Reviewer #3: **Yes: **HOSAM ALRAHHAL

Reviewer #4: No

Reviewer #5: No